# Catalytic Cracking of Biodiesel Waste Using Metal Supported SBA-15 Mesoporous Catalysts

**Duangkamon Jiraroj [1], Tunyatorn Tongtooltush [2], Joongjai Panpranot [3], Piyasan Praserthdam [3] and Duangamol Nuntasri Tungasmita [1,*]**

[1] Center of Excellence in Catalysis for Bioenergy and Renewable Chemicals (CBRC), Department of Chemistry, Faculty of Science, Chulalongkorn University, Bangkok 10330, Thailand; jiraroj_d@hotmail.com

[2] Program of Petrochemistry and Polymer Science, Faculty of Science, Chulalongkorn University, Bangkok 10330, Thailand; nrotaynut@hotmail.com

[3] Department of Chemical Engineering, Faculty of Engineering, Chulalongkorn University, Bangkok 10330, Thailand. joongjai.p@chula.ac.th (J.P.); piyasan.p@chula.ac.th (P.P.)

* Correspondence: duangamol.n@chula.ac.th; Tel.: +66-2-218-7619

**Abstract:** Palladium (Pd) and aluminium (Al) supported on SBA-15 were prepared as catalysts for cracking biodiesel waste from biodiesel production. Mesoporous silica SBA-15 was first synthesized by a hydrothermal method and then loaded with Al or Pd particles were loaded using postsynthesis or aqueous wet impregnation methods, respectively. The physical properties of the catalysts were characterized by X-ray diffraction (XRD), nitrogen ($N_2$) adsorption, scanning electron microscopy (SEM), and transmission electron microscopy (TEM) analyses. The catalytic cracking performance of biodiesel waste was evaluated at reaction temperatures above 400 °C under a $N_2$ atmosphere in a batch reactor for 40 min in comparison with that for pure glycerol, where the conversion of biodiesel waste reached 86.8% with 10 wt% Pd-SBA-15 at 650 °C. The product types depended on whether the starting material was pure glycerol or biodiesel waste. The main gaseous products were carbon monoxide as synthesis gas, carbon dioxide, and 1,3-butadiene. Additionally, 2-cyclopenten-1-one and 2-propen-1-ol were major products in the liquid fraction, which can be used in pharmaceuticals and as a flame retardant, respectively.

**Keywords:** glycerol cracking; biodiesel waste; Al–SBA-15; Pd–SBA-15; mesoporous material

## 1. Introduction

The world is currently being confronted with an energy crisis, due to the depletion of resources, especially nonrenewable petroleum, and increasing environmental problems. Biodiesel plays an important role as a new alternative energy source that is widely recognized as a potential partial solution to these energy problems, and so the demand for biodiesel has increased rapidly. During biodiesel processing, crude glycerol is formed as the main by-product, but it is impure and of low economic value, and so poses a waste problem. There are several available methods for managing biodiesel by-products, such as separation, purification, and pyrolysis.

Glycerol, a polyol molecule containing three hydroxyl groups, is an important chemical reactant for various applications, such as in the food, cosmetic and pharmaceutical industries [1,2]. However, glycerol can also be converted to valuable products as hydrocarbon molecules and synthesis gas [3–7]. Techniques used for the transformation of glycerol include hydrogenolysis [8–12], dehydration [13–16], esterification [17,18], carboxylation [19], catalytic cracking [20,21], and other reactions [22,23]. Pyrolysis can transform biodiesel waste to more useful chemicals, but it exhibits low conversion level and selectivity. In contrast, catalytic cracking can achieve higher activity and product selectivity. Various

types of catalysts have been used in glycerol transformation, such as ZSM-5 [14], metal/ZSM-5 [14], cerium (Ce)-nickel (Ni)/alumina ($\alpha$Al$_2$O$_3$) [21], MCM-41 [20], Ce-Ni-SBA-15 [9], and others [24,25]. Previous research has evaluated the pyrolysis of glycerol over a 3% by weight (wt%) Pr-Ni/$\alpha$-Al$_2$O$_3$ catalyst in order to produce syngas, and reported that the main gaseous products were hydrogen (H$_2$), carbon monoxide (CO), carbon dioxide (CO$_2$), and methane (CH$_4$) [3]. The addition of molybdenum oxide (MoO$_3$)-modified Ni$_2$P/Al$_2$O$_3$ incorporated with ZSM-5 was applied as catalyst for the hydrogenolysis of glycerol to propylene, and high product selectivity of ~88% was obtained [10]. However, the high acidity mesoporous ZSM-5 material exhibited the best performance for acrolein formation (70% selectivity) from glycerol dehydration [15]. The combination of ZSM-5 and bentonite was used as a catalyst in the ex situ catalytic pyrolysis of crude glycerol for the synthesis of bio-based benzene, toluene and xylenes [26]. Additionally, activated carbon was investigated as a catalyst for glycerol pyrolysis using microwave heating, where the carbonaceous catalyst improved the H$_2$ selectivity at a high reaction temperature, ensuring a high proportion of gaseous products [6].

However, the use of a noble metallic catalyst, such as rubidium (Rb), palladium (Pd), platinum (Pt), or copper (Cu), supported on a porous material, has been preferred for the transformation of glycerol. Previously, Pd or Pt particles were loaded onto a carbon support to oxidize glycerol to lactic acid [27]. This catalyst showed a good performance compared to the hydrothermal method even at a low temperature with a low glycerol:alkali ratio. In addition, the direct catalytic pyrolysis of pure glycerol over a heterogeneous Ce–Ni/$\alpha$Al$_2$O$_3$ catalyst gave H$_2$ and CO as the main products (synthesis gas) with CH$_4$ and CO$_2$ as by-products [21].

Furthermore, microporous materials, including both supported ZSM-5 and metal-impregnated ZSM-5, were applied for the conversion of glycerol to light olefins via dehydration. The use of a Cu-loaded Cu/ZSM-5 catalyst gave CH$_4$, ethylene (C$_2$H$_4$), CO, and CO$_2$ with high product selectivity and yield [14]. Moreover, when mesoporous SBA-15 catalysts were applied as the supporting material for glycerol conversion and loaded with metallic Ce and Ni to increase the acid sites, volatile products were formed via hydrogenolysis, with 1,2-propanediol as the major product [9]. The use of H$_3$PW$_{12}$O$_{40}$ supported on Cs/SBA-15 was investigated in the dehydration of crude and pure glycerol, where the material was found to be durable at high temperatures and showed a good activity and selectivity for acrolein as the product [25,28]. Finally, tungsten oxide was loaded onto Zr-SBA-15 as catalyst for the glycerol dehydration to acrolein [24].

Although several studies have been conducted on the conversion of glycerol under different conditions, only a few studies have reported on mesoporous structures as catalysts in glycerol cracking. Therefore, mesoporous silica SBA-15 was used as a supporting material in this study, since it has a high thermal stability, parallel cylindrical pores of 9–10 nm diameter, and axes arranged in a hexagonal unit cell. In addition, this material has a high specific surface area with a great diversity in potential surface modifications and large pore sizes, allowing for easy diffusion of bulky reactants and products through the porous network during reactions. In addition, the supporting SBA-15 was modified with metallic particles to enhance its catalytic properties for application in glycerol transformation.

Metal oxides have previously been loaded on the SBA-15 structure to form catalysts for glycerol dehydration. For example, the use of cobalt (Co) and Ru supported on SBA-15 achieved the best yields for 1-hydroxyacetone [29], while in the hydrogenolysis of glycerol, Ru/SBA-15 gave CH$_4$ and 1,2-propanediol as the main products [8]. The production of H$_2$ from glycerol steam reforming was investigated over a Ni/SBA-15-supported alkaline catalyst [22].

This research aimed to increase the value of the glycerol containing biodiesel waste by catalytic cracking. The effect of the reaction temperature, the catalyst type (Al- or Pd-coated SBA-15), and the amount of catalyst were investigated in the cracking of biodiesel waste in comparison to that with pure glycerol. The prepared mesoporous catalysts SBA-15, Al-SBA-15, and Pd-SBA-15 were characterized by several techniques. This reaction produced both liquid and gas fractions, the contents of which were determined by gas chromatography (GC). The reaction was efficient with respect to the transformation of glycerol to the valuable products.

## 2. Results and Discussion

### 2.1. Catalyst Characterization

The physical properties of the synthesized materials were characterized by X-ray powder diffraction (XRD), nitrogen ($N_2$) adsorption/desorption isotherms, scanning electron microscopy (SEM), SEM with energy dispersive spectrometry (SEM-EDS), transmission electron microscopy (TEM), and $^{27}$Al nuclear magnetic resonance ($^{27}$Al-NMR). The structures of SBA-15, Al-SBA-15, and Pd-SBA-15 were confirmed by XRD, with the results are shown in Figure 1. The XRD patterns of all of the SBA-15 catalysts were similar, showing three well-resolved peaks that were assigned to the (100), (110), and (200) reflection planes. These corresponded to the well-ordered hexagonal structure of SBA-15. For Al-SBA-15, the peak pattern was slightly shifted to a higher angle because of shrinkage during recalcination or changes in the lattice parameters resulting from the inclusion of Al in the SBA-15 structure, which is consistent with previous research [30,31]. For the Pd-SBA-15 catalyst, the prominent XRD reflection (100) slightly shifted towards a higher angle, which was due to shrinkage during recalcination as the unit cell contracted after heating at a high temperature [22].

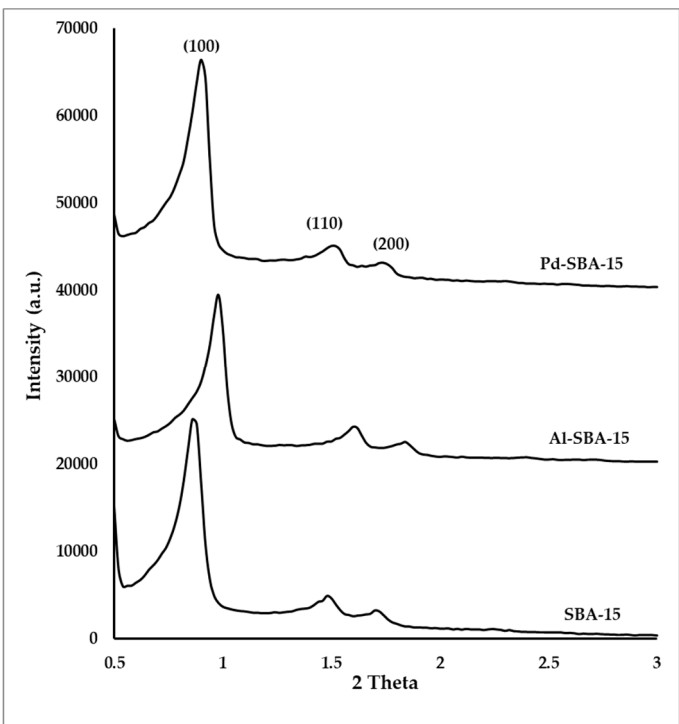

**Figure 1.** Representative X-ray diffraction (XRD) diffractograms of the as-formed SBA-15, Al–SBA-15, and Pd–SBA-15 catalysts.

The physicochemical properties of the prepared SBA-15 catalysts, as obtained from the $N_2$ adsorption/desorption results, are shown in Figure 2 and Table 1. All three catalysts exhibited type IV isotherms, which are typical for mesoporous materials [22,23], and had pore diameters of 9.23 nm. Additionally, loading of the Al on the SBA-15 dramatically decreased (1.97-fold) the specific surface area from 925 to 469 $m^2g^{-1}$, which is consistent with a previous report that the addition of Al to silica catalysts decreased the specific surface area [30]. Furthermore, after metal impregnation, the shapes of the adsorption/desorption isotherms remained similar to those of SBA-15, indicating that the uniform pore structure of SBA-15 was still maintained even after impregnation. The total specific surface area and external surface area of the Pd–SBA-15 catalyst decreased 1.25- and 1.21-fold, respectively, from that for the pure SBA-15 material. Therefore, the results supported that the metal mainly remained on the surface rather than entering the channels.

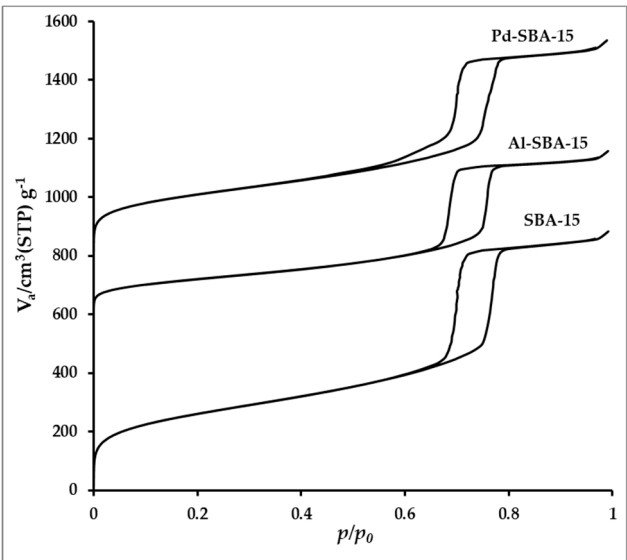

**Figure 2.** Representative N$_2$ adsorption/desorption isotherms and pore distributions of the SBA-15, Al–SBA-15, and Pd–SBA-15 catalysts.

**Table 1.** Physicochemical properties of the SBA-15, Al–SBA-15, and Pd–SBA-15 catalysts.

| Catalyst | Total Specific Surface Area [a] (m$^2 \cdot$g$^{-1}$) | External Surface Area [b] (m$^2 \cdot$g$^{-1}$) | Pore size Distribution [c] (nm) | Pore Volume [c] (cm$^3 \cdot$g$^{-1}$) | $d_{(100)}$ [d] (nm) |
|---|---|---|---|---|---|
| SBA-15 | 925 | 51 | 9.23 | 1.24 | 9.69 |
| Al–SBA-15 | 469 | 49 | 9.23 | 0.90 | 9.31 |
| Pd–SBA-15 | 740 | 42 | 9.23 | 1.05 | 10.10 |

[a] Calculated by the [a] BET, [b] t-plot, and [c] BJH methods or [d] from the XRD analysis using the Jade 5.6 software.

The morphology and structure of SBA-15 and the three different metal-loaded SBA-15 catalysts were studied using SEM and TEM analyses, respectively, with representative images shown in Figure 3. All samples formed aggregated particles with rod-like structures of an average size of around 0.8 × 1.1 µm. In addition, the agglomerated structure of SBA-15 was separated into small rod-shaped particles after the addition of Al. However, the rod shape of individual particles remained the same in Al-SBA-15 as for the pure SBA-15. After metal addition, the small rod-shaped particles were slightly separated from the rope-like agglomeration. Additionally, the SEM-EDS analysis gave a Pd loading of 2.85% on the SBA-15 support. That the value obtained by SEM-EDS analysis was lower than the metal solution loading (5 wt%) likely reflects that some metals remained in the solution.

In addition, the TEM images confirmed the microstructure of SBA-15, which presented well-ordered hexagonal arrays of one-dimensional mesoporous channels. Generally, Al loaded into the porous materials formed tetrahedral sites that were responsible for the acidity of Al-SBA-15, thus the TEM analysis could not confirm the presence of the Al particles. The location of Al atoms on the support material was therefore investigated by $^{27}$Al-MAS-NMR analysis. Only the signal of the framework site at 50 ppm was observed, which was ascribed to the Al atoms incorporated into tetrahedral framework positions. For Pd-SBA-15, the TEM image confirmed the location of metallic particles outside the channels of the SBA-15 structure, with sizes of ~11 nm. However, some Pd particles could not be accommodated within the channels of the support was because their size significantly exceeded the pore width of SBA-15 (estimated by the BJH method to be 9.23 nm). These results suggested that the impregnated Pd was loaded on the external surface of the catalyst, where a minority of the silanol groups were located, rather than into the pores [32].

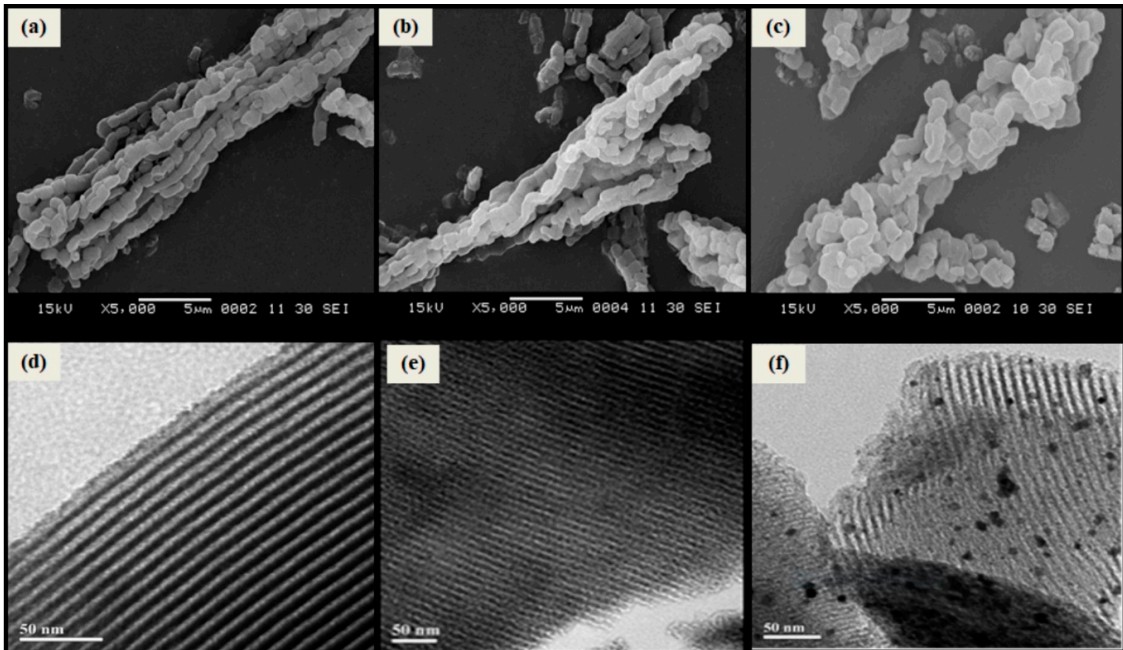

**Figure 3.** Representative (**a**–**c**) SEM (5000× magnification) and (**d**–**f**) TEM (300,000× magnification) micrographs of SBA-15 (**a,d**), Al–SBA-15 (**b,e**), and Pd–SBA-15 (**c,f**).

## 2.2. Catalytic Performance

The composition of the biodiesel waste was determined by using a standard method [4–7] in order to predict the cracking products. The analysis revealed a 54.5% matter organic non-glycerol (MONG) content, which consisted of mainly methyl ester molecules, 37.2% glycerol, 6.5% ash, and 1.8% water content. On the other hand, the composition of the pure glycerol was 1.7%, 98.2%, and 0.1% for MONG, glycerol, and ash, respectively, with no detected water content. In addition, the density of the pure glycerol and biodiesel waste were slightly different at 1.26 and 1.03 g/mL at 20 °C, respectively, while the pH was significantly different being acidic (pH 5.17) for the pure glycerol and alkaline (pH 10.47) for the biodiesel waste (reflecting the residual alkali catalyst used in the transesterification reaction in the biodiesel production). Thus, the biodiesel waste could act as a basic reagent. The % conversion and % product yield for the cracking of biodiesel waste and pure glycerol over SBA-15, Al–SBA-15, and Pd–SBA-15 catalysts were determined, with the results summarized and shown in Table 2. Both thermal cracking (no catalyst) and catalytic cracking were investigated with both the pure glycerol and biodiesel waste as the starting material at reaction temperatures of 400, 650, and 800 °C for 40 min.

**Table 2.** Conversion and product selectivity for the thermal and catalytic cracking of glycerol and biodiesel waste.

| Catalysts | Temp. (°C) | Catalyst Amount (wt%) | Starting Material | % Conversion | % Product Selectivity | | | | |
|---|---|---|---|---|---|---|---|---|---|
| | | | | | Gas fraction | Liquid fraction | Distilled Liquid | Heavy Liquid | Residue |
| Thermal cracking | 400 | - | Pure glycerol | 87.8 | 4.0 | 83.8 | 7.8 | 76.0 | 12.2 |
| Thermal cracking | 400 | - | Biodiesel waste | 63.5 | 14.5 | 49.0 | 22.4 | 26.6 | 36.5 |
| Thermal cracking | 650 | - | Biodiesel waste | 85.0 | 28.2 | 56.7 | 23.3 | 33.5 | 15.1 |
| Thermal cracking | 800 | - | Biodiesel waste | 86.3 | 27.3 | 59.0 | 23.8 | 35.2 | 13.7 |
| SBA-15 | 400 | 10 | Biodiesel waste | 66.5 | 15.9 | 50.6 | 15.1 | 35.5 | 42.5 |
| Al–SBA-15 | 400 | 10 | Pure glycerol | 89.6 | 5.7 | 83.9 | 9.0 | 74.9 | 10.4 |
| Al–SBA-15 | 400 | 10 | Biodiesel waste | 65.5 | 17.5 | 48.0 | 18.8 | 29.2 | 34.5 |
| Al–SBA-15 | 650 | 10 | Biodiesel waste | 86.8 | 25.6 | 61.2 | 25.8 | 35.4 | 13.2 |
| Pd–SBA-15 | 400 | 10 | Pure glycerol | 93.4 | 14.1 | 79.3 | 12.9 | 66.4 | 6.6 |
| Pd–SBA-15 | 400 | 10 | Biodiesel waste | 69.5 | 25.1 | 45.4 | 15.1 | 30.3 | 29.5 |
| Pd–SBA-15 | 650 | 5 | Biodiesel waste | 85.7 | 26.4 | 59.3 | 20.3 | 39.0 | 14.3 |
| Pd–SBA-15 | 650 | 10 | Biodiesel waste | 86.8 | 33.2 | 53.6 | 18.3 | 35.3 | 13.2 |
| Pd–SBA-15 | 650 | 15 | Biodiesel waste | 86.9 | 32.2 | 54.7 | 18.9 | 35.9 | 13.1 |

### 2.2.1. Effect of the Starting Material

The catalytic activity was initially investigated at 400 °C for 40 min using pure glycerol or the biodiesel waste as the substrate in comparison with the thermal (no catalyst) reaction. The conversion levels and product yields are shown in Table 2 and Figure 4. The catalytic reactions (SBA-15, Al-SBA-15, and Pd-SBA-15) gave slightly higher conversion levels than thermal cracking with pure glycerol or biodiesel waste, respectively, attaining 89.6–93.4% and 65.5–69.5% conversion levels, respectively, compared to 87.8% and 63.5% for the thermal cracking, respectively. Moreover, the presence of the respective catalyst gave a higher yield for both liquid and gaseous products, and so the catalysts increased the reaction efficiency. Comparison between the pure support (SBA-15) and the metal-doped (Al-SBA-15 and Pd-SBA-15) catalysts at 10 wt% revealed that the metal-doped materials increased the gaseous fraction and decreased the residue from the pyrolysis process with both glycerol and biodiesel waste as substrates, and this was especially the case with Pd-SBA-15. Indeed, Pd-SBA-15 also gave the highest conversion level of 93% and 70% for pure glycerol and biodiesel waste, respectively. Overall, the combination of metal particles and mesoporous SBA-15 enhanced the catalytic activity and increased the efficiency of the cracking reaction.

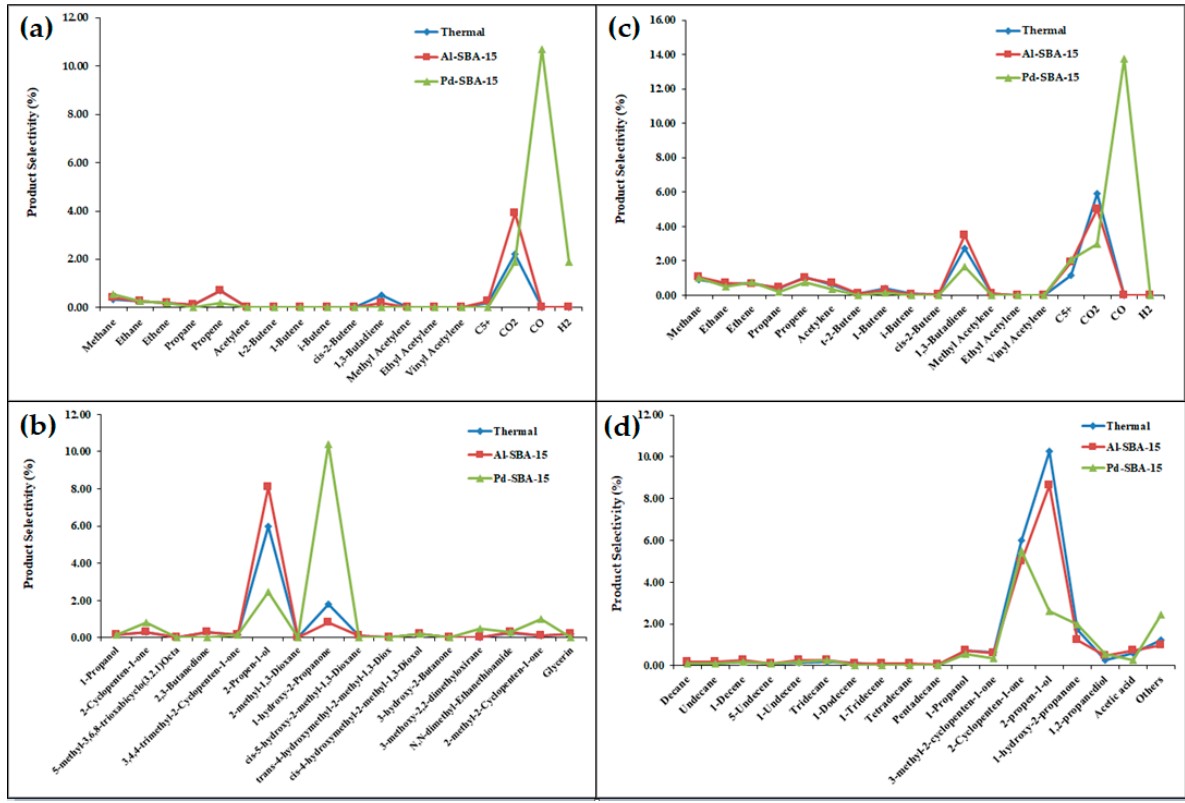

**Figure 4.** Product selectivity of the (**a**,**c**) gas and (**b**,**d**) distilled liquid products from the thermal and catalytic cracking of (**a**,**b**) pure glycerol and (**c**,**d**) biodiesel waste as starting materials at 400 °C. Data are shown as the mean ± 0.2 SD, derived from three independent trials.

Additionally, with either the pure glycerol or the biodiesel waste as the substrate the main liquid fraction product was heavy oil, which is consistent with previous research where metals, such as Co and Ru, supported on SBA-15 catalysts were selective for liquid products in the dehydration reaction forming 1-hydroxyacetone, propanediols, acetic acid, acetone and acrolein as the major products [29]. Moreover, the results of this study showed that the product yield of the gas fraction increased when the biodiesel waste was used as the starting material. The liquid product selectivity reached 84% and 61% for the pure glycerol and biodiesel waste, respectively, which reflects the higher glycerol content in the pure glycerol than in the biodiesel waste (98.2% vs. 37.2%).

For the gas fraction from pure glycerol (Figure 4), the distribution of products was selective for $CO_2$, except with Pd-SBA-15 that showed the highest amount of CO at around 11% along with $CO_2$ and $H_2$, similar to synthesis gas. The biodiesel waste was converted to $CO_2$, 1,3-butadiene, $C_5^+$ and propene, but the Pd-SBA-15 catalyst still gave CO as the major gas product at around 14% (Figure 4c), which is from the decarbonylation of glycerol and MONG [26]. In this study, the distribution of the volatile liquid phases was selective for 1-hydroxy-2-propanone and 2-propen-1-ol for all of the catalysts when using the pure glycerol. When the reaction temperature was above the boiling point of glycerol (290 °C), the liquid fraction was transformed to 1-hydroxy-2-propanone by dehydration [33–35]. This was then reduced and dehydrated to 2-propen-1-ol, as shown in Scheme 1 for the transformation pathway of glycerol into liquid products. Usually, the harvested products in the liquid fraction are mainly produced via dehydration followed by H-transfer or aldol condensation [34,35]. Therefore, several researches have tried to reduce the gas phase production in order to enhance the yield of desired liquid compounds. The association of ZSM-5 and bentonite catalysts could decrease the production of the gaseous fraction via decarboxylation and decarbonylation reactions, giving a high yield of the liquid fraction [26]. As seen in this work, both types of products were observed and the product amount depended on the type of the starting material. With the biodiesel waste material, thermal and Al-SBA-15 cracking gave a higher content of 2-propen-1-ol than 2-cyclopenten-1-one. In contrast, the Pd-SBA-15 catalyst gave 2-cyclopenten-1-one as the major product, which was formed from reacting with acetaldehyde, the catalytic cracking product of MONG, and by aldol condensation.

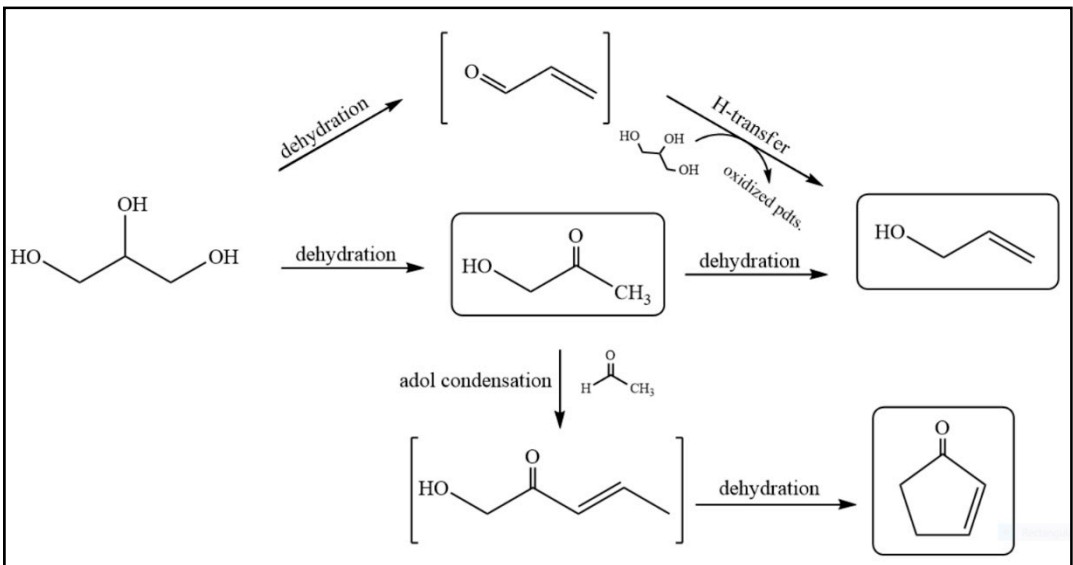

**Scheme 1.** The transformation of glycerol into liquid products.

Since the pure glycerol cracking reaction obtained a high conversion level but only a heavy liquid product, then the biodiesel waste was selected to study the other parameters of the cracking reaction.

### 2.2.2. Effect of the Reaction Temperature without a Catalyst (Thermal Cracking)

The biodiesel waste was reacted at 400, 650, and 800 °C for 40 min without a catalyst and the conversion level and product selectivity of the thermal cracking reactions are shown in Table 2 and Figure 5. The conversion level significantly increased from 63.5% to 85.0% when the reaction temperature was increased from 400 to 650 °C, and then slightly increased to 86.3% as the temperature was increased further to 800 °C. Likewise the residue level decreased dramatically and then slightly with increasing reaction temperature from 400 to 650 and 800 °C. At the low temperature (400 °C), the gaseous products were $CO_2$ and 1,3-butadiene. Increasing the temperature to 650 °C increased the gas fraction selectivity from 14.5% to 28.2% and promoted the production of $CO_2$, $CH_4$, 1,3-butadiene,

$C_2H_4$, $C_3H_6$, and $C_2H_6$, since the higher temperature caused the products to be cracked to smaller molecules. This trend reflected that reported in a previous study, where the yield of gaseous products increased as the glycerol pyrolysis temperature was increased from 400 °C to 800 °C [6]. Likewise, as the temperature increased, smaller gas molecules—$CH_4$ and $C_2H_4$—were obtained, which is again consistent with other experimental data [7]. In addition, the selectivity of liquid products increased to 59%, which was comprised of an increase in heavy oil to ~35% without any significant change in the level of light oil (remained at ~22–24%), and mainly consisted of 2-propen-1-ol and 2-cyclopenten-1-one. Therefore, although 800 °C was likely to be an acceptable reaction temperature, this would result in a high energy consumption and so a temperature of 650 °C was considered as more suitable for the cracking process because a high yield of gas and liquid products were still obtained over the Pd-SBA-15 or Al-SBA-15 catalysts at this temperature, respectively.

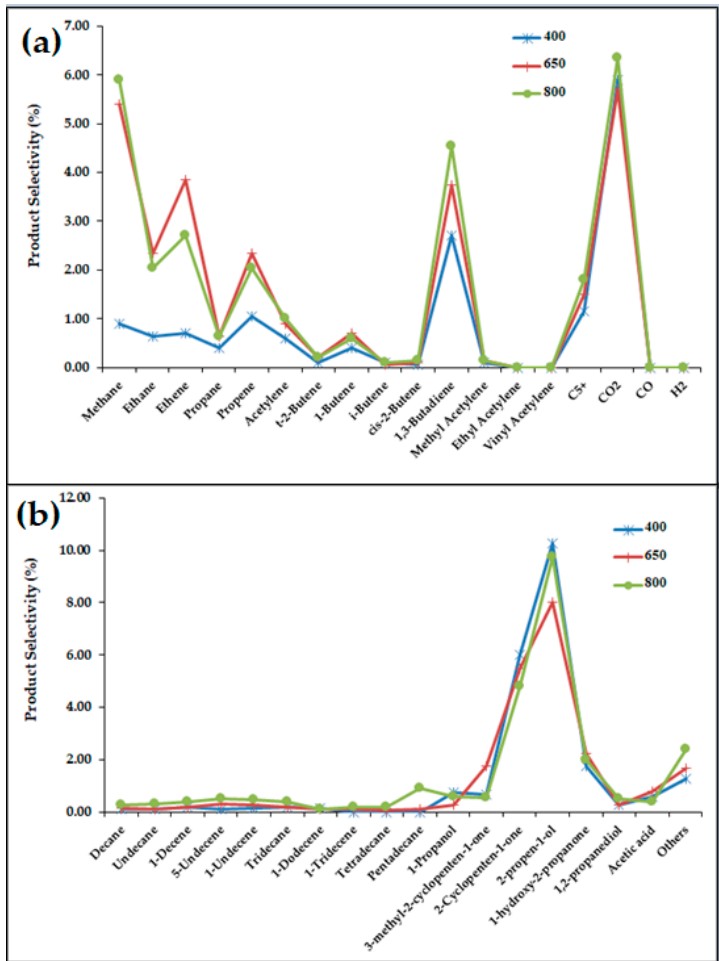

**Figure 5.** Product selectivity for the (**a**) gas and (**b**) distilled liquid products from the thermal cracking of biodiesel waste at different reaction temperatures. Data are shown as the mean ± 0.2SD, derived from three independent trials.

## 2.2.3. Catalytic Cracking of Biodiesel Waste

The effect of the catalyst type was compared for the Al-SBA-15 and Pd-SBA-15 catalysts in biodiesel waste cracking at 650 °C for 40 min. At this temperature, the catalysts did not show a significant difference in the obtained conversion level (~87%), as shown in Table 2. All of the reactions produced a higher selectivity for liquid than gas fractions. With respect to the heavy liquid fraction, Al-SBA-15 gave a higher selectivity for distilled liquid products (26%) than the others, while the Pd-SBA-15 catalyst increased the selectivity for the gas fraction to 33%. The product selectivities of

this cracking reaction are shown in Figure 6. In the distilled liquid fraction, the main products were 2-cyclopenten-1-one and 2-propen-1-ol for both catalysts. Previously, several product types were obtained from glycerol transformation depending on the supported metallic species, where Co or Ru supported on SBA-15 gave 1-hydroxyacetone and acetone as the products, whereas propanediol and acetic acid were produced with Zn- or Cu-SBA-15 catalysts [29].

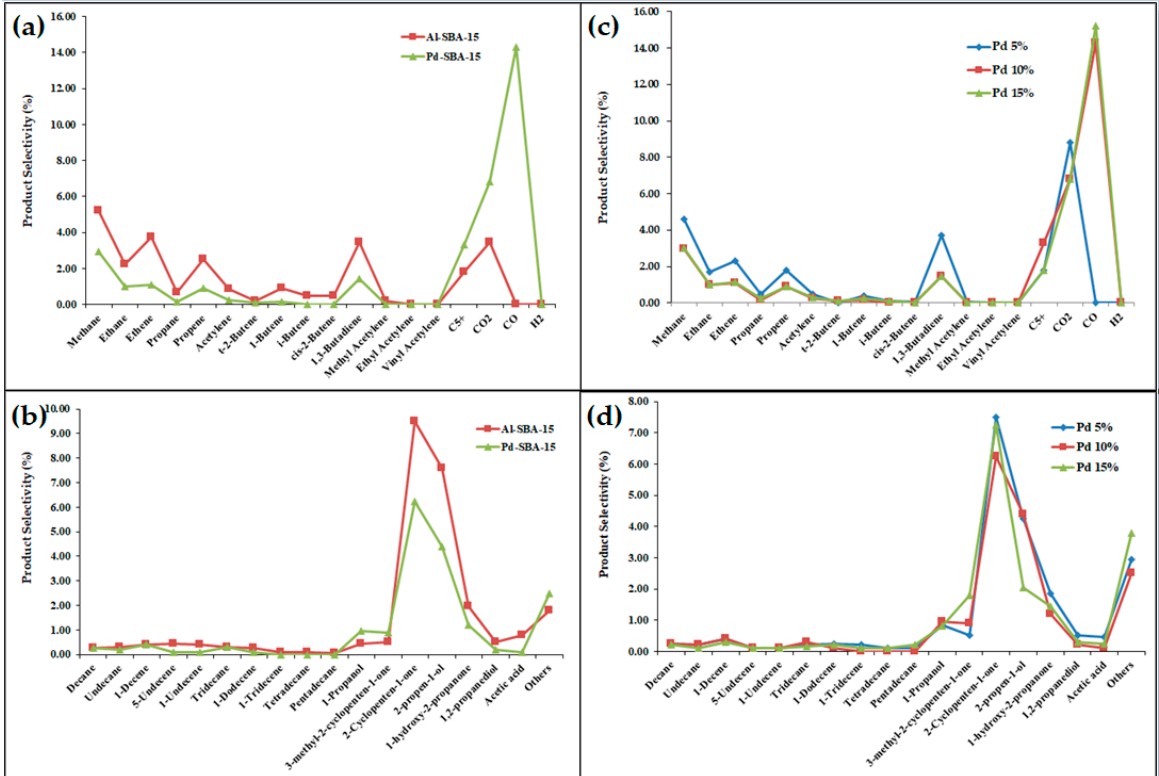

**Figure 6.** Product selectivity for the (**a**,**c**) gas and (**b**,**d**) distilled liquid products from biodiesel waste cracking at 650 °C with different catalyst (**a**,**b**) types and (**c**,**d**) amounts of catalyst. Data are shown as the mean ± 0.2 SD, derived from three independent trials.

For the gas fraction, catalytic pyrolysis with the Al-SBA-15 catalyst provided several gas products, such as $CH_4$, $C_2H_4$, 1,3-butadiene, $CO_2$, propene, and $C_5^+$. Meanwhile, that with the Pd-SBA-15 catalyst gave CO, $CO_2$, and $C_5^+$ as the major gas composition, which are components of synthesis gas. This exhibited a good catalytic activity for the production of CO, a synthesis gas component, with a high CO selectivity above 14% at 650 °C. Therefore, catalytic cracking of biodiesel waste with Pd-SBA-15 at 650 °C was chosen to study the effect of the amount of catalyst, since this gave the highest yield of the gas fraction.

### 2.2.4. Effect of the Amount of Catalyst

The amount of Pd-SBA-15 varied (5, 10, and 15 wt%) in the biodiesel waste cracking reaction at 650 °C for 40 min. Conversion levels of ~85.7–86.9% were observed, and so were not dependent on the amount of catalyst (Table 2). As the amount of Pd-SBA-15 was increased from 5 wt% to 10 or 15 wt%, some liquid products were cracked to gaseous products, increasing the gas fraction by ~7% (from 26.4% to 32.2%), but the conversion level did not change significantly and the amount of residue only marginally decreased as the amount of catalyst was increased. The gaseous and liquid product distributions are shown in Figure 6. The favored gaseous products were CO, $CO_2$, $CH_4$, $C_5^+$, and 1,3-butadiene for reactions with 10–15 wt% Pd-SBA-15, but was different with 5 wt% Pd-SBA-15 where the gaseous products comprised 26% of the total yield, and the reaction was selective

for $CO_2$, 1,3-butadiene, $CH_4$, $C_2H_4$, and propene. Both 10 and 15 wt% Pd-SBA-15 resulted in similar conversions and product yields with CO as the major product at ~15%, which was not produced with 5 wt% Pd-SBA-15. This is consistent with a previous work, in which catalysis of the glycerol steam reforming with a 2.5 wt% $Pd/Al_2O_3$ catalyst resulted in CO as the main gas product [33].

Furthermore, the selectivity of the distilled liquid fraction obtained in this study was not significantly changed for any of the reactions. The favored products were 2-cyclopenten-1-one and 2-propen-1-ol, which can be used as a fragrance and a flame retardant, respectively. Thus, from all of the results, the 10 wt% Pd-SBA-15 catalyst was deemed to be the most suitable for the biodiesel waste cracking reaction.

## 3. Materials and Methods

### 3.1. Preparation of Catalysts

The SBA-15-supported materials were synthesized by a hydrothermal method as reported [36] using the triblock copolymer pluronic P123 and poly(ethylene glycol)-poly(propylene glycol)-poly(ethylene glycol) (Aldrich, $EO_{20}PO_{70}EO_{20}$, average molecular weight 5800), as a template, which was dissolved in 2 M hydrochloric acid (HCl). Tetraethyl orthosilicate (Fluka, TEOS), used as the silica source, was then added and stirred for 1 h giving a gel with a 1: 0.0165: 5.88: 192 molar ratio TEOS: P123: HCl: $H_2O$ composition. This silicate fluid gel was thoroughly stirred at 40 °C for 24 h and then transferred to a Teflon-lined autoclave for hydrothermal crystallization at 100 °C for 48 h. After crystallization, the autoclave was quenched with cold water, and the solid was separated from the mother liquor. The recovered solids were washed with deionized water to pH 7, dried at 100 °C overnight, and calcined at 550 °C for 5 h.

Aluminum particles were loaded onto the SBA-15 catalyst using a postsynthesis impregnation method [37] at a $SiO_2/Al_2O_3$ molar ratio of 10:1, and denoted as Al-SBA-15. In this procedure, 0.5 g of mesoporous SBA-15 was stirred in 50 mL of 30 mM sodium aluminate (Riedel-deHaën, reagent grade) solution at room temperature for 12 h. The Al-containing SBA-15 material was filtered, washed, and dried at 70 °C overnight. After post-treatment, the remaining sodium ions were removed using ion-exchange with 0.01 M ammonium chloride solution. Finally, the prepared material was calcined at 550 °C for 5 h.

Palladium supported on siliceous SBA-15 was prepared by aqueous wet impregnation [38] using palladium chloride (Aldrich) dissolved in 1 M HCl at a metal loading of 5 wt%. After impregnation, the material was dried at 120 °C and then calcined at 500 °C for 4 h. The synthesized material was denoted as Pd-SBA-15.

### 3.2. Catalyst Characterization

The physical properties of the catalysts were studied as follows. The structure of the materials was determined by XRD analysis (Rigaku D/MAX-2200, Ultima-plus, Tokyo, Japan) with graphite monochromatized Cu K$\alpha$ radiation at 40 kV and 30 mA. Samples were scanned between 0.5° and 3.0° at a scan speed of 1.0 min. The scattering, divergent, and receiving slits were fixed at 0.5°, 0.5°, and 0.15 mm, respectively. The $N_2$ adsorption–desorption isotherms were obtained at 77 K using a conventional volumetric apparatus (BEL Japan, BELSORP-mini, Osaka, Japan). Prior to the adsorption measurements, the calcined samples (around 40 mg) were evacuated at 400 °C for 3 h. The specific surface area and pore size distribution of the solids were evaluated by the Brunauer–Emmett–Teller (BET) method and the Barrett–Joyner–Halenda (BJH) method, respectively.

The morphology of the crystallites was determined by SEM (JEOL JSM-5410 LV and JEOL JSM-6480 LV, Peabody, MA, USA) using a scanning electron microscope with an acceleration voltage of 15 kV. Additionally, SEM-EDS was used to measure the metal loading in the support material. For TEM (JEOL; JEM-2100, Peabody, MA, USA), solid samples were prepared by dispersing the synthesized powder in ethanol.

### 3.3. Catalytic Activity Test

The glycerol cracking was investigated using pure glycerol (Fluka, 99.5%) and biodiesel waste from a biodiesel production plant as the starting material. The cracking was performed in a stainless steel tubular reactor at various temperatures (400–800 °C) and a heating rate of 20 °C/min at atmospheric pressure with a 20 mL/min $N_2$ flow. The apparatus was equipped with an electronic temperature controller for a furnace. The temperature was measured and controlled using a K-type thermocouple set at the axial center of the tubular reactor (Figure 7). In each cycle, 5.0 g of glycerol sample was loaded into the reactor and mixed thoroughly with 10 wt% of catalyst. Afterwards, the reactor was purged with $N_2$ at a flow rate of 20 $cm^3$/min. Once the desired temperature was reached (400, 650, or 800 °C), it was kept constant for 40 min. After completion of the reaction, the reactor was cooled to room temperature and weighed. The cracking product was condensed and separated into the three main fractions of liquid, gas, and solid residues.

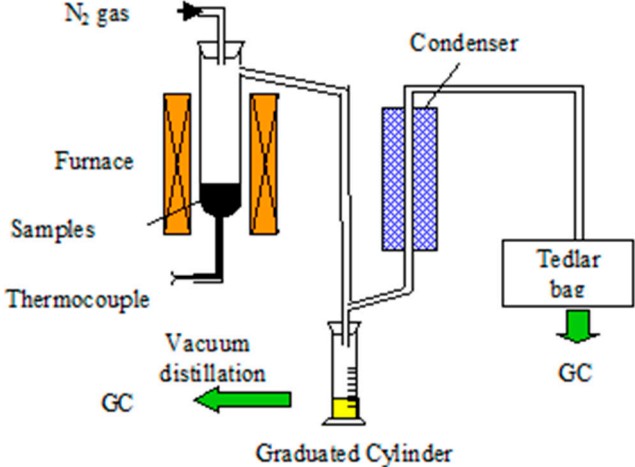

**Figure 7.** Schematic diagram of the apparatus used for the thermal or catalytic cracking process [30].

The gas fraction was collected in a 3-L Tedlar bag from the onset of heating, and the liquid fraction was collected in a 10-$cm^3$ graduated cylinder after condensation with water at 10 °C. The % conversion level was defined as the sum of the collected gaseous and liquid products with respect to the amount of initially loaded starting material. The solid remaining in the reactor was considered as the residue and was not included in the conversion. The liquid fraction obtained from the reaction was separated into light and heavy liquid fractions by vacuum distillation at 200 °C. The solid residue referred to the carbonaceous solids and the waxy compounds that remained in the line of the catalytic reactor. The cracking process was performed three times in order to confirm the reproducibility, with the results being within ± 0.2% of the standard deviation (SD).

The cracking products were analyzed by GC using a Varian CP-3800 gas chromatograph and an Alumina-PLOT column (50 m length × 0.53 mm O.D.) connected to a flame ionization detector for hydrocarbon analysis. A ShinCarbon ST micropacked column (2 m, i.d. 1 mm) was connected to a thermal conductivity detector, in which permanent gases were analyzed.

$$\% \text{ Conversion } = \frac{\text{Mass of liquid products} + \text{mass of gaseous products}}{\text{Mass of starting plastic material}} \times 100$$

mass of gas fraction = mass of the reactor with plastic and catalyst before reaction − mass of the reactor with residue and used catalyst after reaction − mass of liquid fraction

$$\% \text{ Yield } = \frac{\text{Mass of product fraction}}{\text{Mass of starting plastic material}} \times 100$$

## 4. Conclusions

Aluminium- or Pd-supported SBA-15 catalysts were successfully prepared by postsynthesis and wet impregnation methods, respectively, as Al-SBA-15 and Pd-SBA-15. The structure of these synthesized catalysts remained mesoporous, similar to the supporting SBA-15 material. During metal loading, Al atoms entered the porous framework, whereas Pd particles remained on the surface with a metallic particle size around 11 nm. Catalytic cracking of glycerol gave a higher conversion level than thermal cracking. Comparison of the catalytic cracking with pure glycerol or biodiesel waste as the substrate at 400 °C revealed that pure glycerol provided a higher level of liquid products (up to 84%), which were mainly composed of heavy oils, than biodiesel waste (45–51%), but catalytic cracking of the biodiesel waste gave a larger gas fraction (around 12%) than with the pure glycerol. At a higher temperature (650 °C), the conversion level increased and the residue level decreased. The more suitable condition for the catalytic cracking of biodiesel waste was deemed to be 10 wt% catalyst at 650 °C for 40 min. The Pd-SBA-15 catalyst gave a higher gaseous product (33%) and selectivity for CO, whereas Al-SBA-15 gave $CO_2$, 1,3-butadiene, $CH_4$, and $C_2H_4$ as the major gas products. With respect to the liquid fraction, both materials provided a similar and high selectivity for 2-propen-1-ol and 2-cyclopenten-1-one. Therefore, the product type depended on the starting material, reaction temperature and catalytic type.

**Author Contributions:** Conceptualization and Methodology, D.J., T.T., J.P., P.P., and D.N.T.; Investigation, D.J., T.T., and D.N.T.; Writing—Original Draft Preparation, D.J. and D.N.T.; Writing—Review & Editing, J.P., P.P., and D.N.T.

**Funding:** This research was funded by the Thailand Research Fund for BRG6180001 and CAT-REAC industrial project RDG6150012 and the Ratchadapisek Somphot Fund for Postdoctoral Fellowship, Chulalongkorn University.

**Conflicts of Interest:** The authors declare no conflicts of interest.

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
