# Peer review of "Catalytic Cracking of Biodiesel Waste Using Metal Supported SBA-15 Mesoporous Catalysts"

_catalysts, doi:10.3390/catal9030291_

Round 1

Reviewer 1 Report

The work Jiraroj and coworkers delves with the glycerol upgrading using a Pt supported on SBA-15 catalyst. Although the topic fits within the scope of the journal and more in particular within the scope of the special issue, there are mayor hurdles that lead me to conclude that the publication of this work in Catalysts is not recommend. My decision is based on troubles at experimental level, discussion level and scientific-soundness level:

§ Experimental level: The experimental setup used in this work for the upgrading (Figure 7) is unable to deliver a proper contact between the reactants and the catalyst and care should be taken with the results presented. A proper fixed bed reactor would be far better, not to mention a fluidized bed. With the system used, non-volatile recalcitrant products would with be in contact with catalyst and polymerize, as demonstrated in the massive residue formation shown in Table 2. On the other hand, the composition of gases demonstrate that conversion is controlled by decarbonylation and decarboxylation (unwanted reactions). 

§ Discussion level: Somehow linked with the previous level. The characterization of reactants and products are extremely poor. Thus, it is impossible to rationalize faithful mechanistic or catalytic interpretation on the grounds of the results presented. Scheme 1 is coming out of the blue and it is not supported by the data presented in this work. Besides, this scheme fails to explain the complex product distribution observed. Additionally, this scheme does not include the truly observed reactions as decarbonylation and decarboxylation. 

§ Scientific soundness level: The English used has very poor style. The discussions do not deep into the latest challenges and the work leaves a taste of “cumulative”. The terminology used for biodiesel waste or glycerol waste is vague, and this is crucial to understand the message of the work. The catalyst is not a metal mesoporous… one but a Pt supported on mesoporous material one. Throughout the text many other inconsistencies and vague statements are observed, as for example numbers of the axes in Figure 2.

Author Response

I'm really appreciated your comments and suggestion, the answer for each question was following and with attachment.

The work Jiraroj and coworkers delves with the glycerol upgrading using a Pt supported on SBA-15 catalyst. Although the topic fits within the scope of the journal and more in particular within the scope of the special issue, there are mayor hurdles that lead me to conclude that the publication of this work in Catalysts is not recommend. My decision is based on troubles at experimental level, discussion level and scientific-soundness level:

§ Experimental level: The experimental setup used in this work for the upgrading (Figure 7) is unable to deliver a proper contact between the reactants and the catalyst and care should be taken with the results presented. A proper fixed bed reactor would be far better, not to mention a fluidized bed. With the system used, non-volatile recalcitrant products would with be in contact with catalyst and polymerize, as demonstrated in the massive residue formation shown in Table 2. On the other hand, the composition of gases demonstrate that conversion is controlled by decarbonylation and decarboxylation (unwanted reactions). 

Answer:

- In the experimental part, the effect of catalyst position was studied with10 wt% Pd-SBA-15 catalyst at 650⁰C for 40 min. Two different catalyst positions were compared normally in liquid-phase contact (catalyst was mixed together with starting material at the bottom of a reactor) and vapor-phase contact (catalyst was placed on the 150-mesh sieve at the center of the reactor).

The results showed that conversion and yield value were slightly different. In the liquid-phase position provided higher conversion and gas fraction value. Additionally, the residue on catalyst was decreased with liquid-phase contact. Moreover, both positions gave similar gas and liquid main products. Thus, this work liquid-phase contact was adopted in the reaction testing system.

§ Discussion level: Somehow linked with the previous level. The characterization of reactants and products are extremely poor. Thus, it is impossible to rationalize faithful mechanistic or catalytic interpretation on the grounds of the results presented. Scheme 1 is coming out of the blue and it is not supported by the data presented in this work. Besides, this scheme fails to explain the complex product distribution observed. Additionally, this scheme does not include the truly observed reactions as decarbonylation and decarboxylation. 

Answer:

- From product identification, GC-MS technique was utilized to identify for liquid products that all product structures were matched well with their MS spectra. For the cracking reaction can produce both liquid and gas. Liquid fraction was produced via dehydration mostly following aldol condensation or H-transfer as shown in Scheme 1. Therefore, the major products of liquid were 2-propen-1-ol and 2-cyclopenten-1-one. The transformation of glycerol to liquid products was concordant with previous studies. The hydroxyacetone and acrolein are both products of the dehydration of glycerol, and allyl alcohol could be obtained either through the direct hydrogenation of acrolein with molecular hydrogen, or through a hydrogen transfer reaction with an alcohol (glycerol or some intermediates in this case) as hydrogen donor. The final product was 2-propen-1-ol. The proposed reaction mechanism was corresponded with ref 34 as shown below and reaction details were explained in manuscript line 199-212.

§ Scientific soundness level: The English used has very poor style. The discussions do not deep into the latest challenges and the work leaves a taste of “cumulative”. The terminology used for biodiesel waste or glycerol waste is vague, and this is crucial to understand the message of the work. The catalyst is not a metal mesoporous… one but a Pt supported on mesoporous material one. Throughout the text many other inconsistencies and vague statements are observed, as for example numbers of the axes in Figure 2.

Answer:

- Our manuscript writing style has been edited by a native speaker (Dr. Robert Douglas John Butcher, university research counselor).

- In the manuscript used the biodiesel waste replace to glycerol waste because the sample was obtained from the biodiesel production that consisted of glycerol as a by-product. The composition of biodiesel was also identified and described in catalytic performance part 2.2, line 156-159.

- The words of “metal mesoporous material” in title was replaced with “metal supported mesoporous material” as reviewer recommended.

- All figures were made more consistence with clear.

Reviewer 2 Report

a. The reviewers recommend to try increasing concentration of catalyst beyond 15% with Pd-SBA15 and show the results of increasing or decreasing gas and liquid fractions.

b. it would be great to see a tabular data showing expected cost of obtained product using this method compared to other methods. 

Author Response

Answer to reviewer 2

a. The reviewers recommend to try increasing concentration of catalyst beyond 15% with Pd-SBA15 and show the results of increasing or decreasing gas and liquid fractions.

Answer:

Catalysts

Catalyst

Amount (wt%)

% Conversion

% Product Selectivity

Gas fraction

Liquid fraction

Distilled Liquid

Heavy Liquid

Residue

Pd–SBA-15

5

85.7

26.4

59.3

20.3

39.0

14.3

Pd–SBA-15

10

86.8

33.2

53.6

18.3

35.3

13.2

Pd–SBA-15

15

86.9

32.2

54.7

18.9

35.9

13.1

- The information in table was indicated from Table 2 in the manuscript.

- The results showed that the increasing concentration of catalyst from 10% to 15% was no significant difference in conversion and selectivity. Thus, 10wt% of catalyst amount was chosen for optimum condition.

b. it would be great to see a tabular data showing expected cost of obtained product using this method compared to other methods. 

The gas and liquid products were compared with other literatures as shown below.

Products

Expected cost of   obtained products

methods

In this work   (cracking)

Ref.3 (pyrolysis)

Ref.6 (pyrolysis)

Ref. 9   (hydrogenolysis)

Ref.10 (dehydration)

Ref.15 (dehydration)  

Ref.27 (pyrolysis)

Ref.28 (oxidation)

Liquid products

2-cyclopenten-1-one

120 SGD/5 g

ü

2-propen-1-ol

48.90 SGD/100 ml

ü

1-hydroxy-2-propanone

58 SGD/5 g

ü

ü

1,2-propanediol

68.10 SGD/25 ml

ü

acrolein

52.3 SGD/pk

ü

ü

Benzene

175 SGD/L

ü

toluene

126 SGD/L

ü

xylene

152 SGD/4 L

ü

lactic acid

96.20 SGD/kg

ü

Gaseous products

CO

ü

ü

ü

ü

CO2

ü

ü

ü

ü

C5+

ü

ü

H2

ü

ü

1,3-butadiene (15% in hexane)

387.0 SGD/250 g

ü

propene

127.0 SGD/L

ü

ethene

125.0 SGD/L

ü

ü

ü

ethane

734.0 SGD/110 g

ü

ü

ü

methane

320.0 SGD/52 L

ü

ü

ü

ü

Reviewer 3 Report

The authors investigated thermal and catalytic cracking of glycerol at moderate to high temperatures using Pd, and Al SBA-15 catalysts. They also tested pure and waste glycerol which contained methyl esters and residual base. Their study is very thorough in analysis including average replicated runs, full analysis of gas and liquid products, and characterization of the catalysts. I have only a few suggestions and comments regarding their work.

Did you titrate the pure glycerol to the same pH as waste glycerol to determine the effect of pH in glycerol cracking vs the presence of methanol in waste glycerol?

How did you detect methylated glycols in the waste glycerol? The alkaline environment typically cleaves most ester bonds such as methyl glycerol.

The page number in the upper right hand corner is inconsistent. It has "2 of 14" on page 8.

Thank you for addressing these questions and concerns.

Author Response

Answer to reviewer 3

Did you titrate the pure glycerol to the same pH as waste glycerol to determine the effect of pH in glycerol cracking vs the presence of methanol in waste glycerol?

Answer:

-         In this work, both starting materials were utilized as received without adjust the acidity or basicity.

How did you detect methylated glycols in the waste glycerol? The alkaline environment typically cleaves most ester bonds such as methyl glycerol.

Answer:

-        In the experimental, the methylated glycols might be in the Metter Organic Non-Glycerol (MONG) which detected by subtracting the summation of the quantity of glycerol, ash and water from 100, following as equation;

The page number in the upper right hand corner is inconsistent. It has "2 of 14" on page 8.

Answer: The page number 8 of the manuscript was corrected.

Reviewer 4 Report

The authors of the current manuscript describes the catalytic cracking of biodiesel waste using metal mesoporous SBA-15 catalysts. The paper is interesting, but the authors should choose to revise the paper according the proposed changes, before it is finally published in the journal.

Mayor corrections:

The manuscript require major revisions. The authors should choose to revise the paper thoroughly according the proposed changes:

1.    The manuscript that the authors have presented is very poorly structured. The reorganization of the manuscript is recommend before it can be published. In this sense, the order of the sections must be:

                1.    Introduction,

                2.    Materials and method,

                3.    Results and discussion,

                4.    Conclusions.

 2.    Introduction. It would be very interesting that the authors of the manuscript cite some recent papers in the introduction like the year 2017 and 2018.

 3.    The Figures 3, 4, 5 and 6 should be clear, improve the quality of images.

 4.    The conclusions should be amplified and improved to clarify the results obtained in this work.

Minor corrections:

Although the paper looks essentially as a valuable contribution, still some details need to be fixed, in my opinion:

 1. The English style requires improvements. The authors must be sure that there are no grammatical errors and that the manuscript is perfectly written in English.

2.  The authors must be sure that the bibliographical references, figures and tables are in accordance with the format of the journal.

3. It would be very interesting that the authors of the manuscript include more bibliographical references and, specially, references from the journal itself.

Author Response

Answer to reviewer 4

The authors of the current manuscript describes the catalytic cracking of biodiesel waste using metal mesoporous SBA-15 catalysts. The paper is interesting, but the authors should choose to revise the paper according the proposed changes, before it is finally published in the journal.

Mayor corrections:

The manuscript require major revisions. The authors should choose to revise the paper thoroughly according the proposed changes:

1.    The manuscript that the authors have presented is very poorly structured. The reorganization of the manuscript is recommend before it can be published. In this sense, the order of the sections must be:

                1.    Introduction,

                2.    Materials and method,

                3.    Results and discussion,

                4.    Conclusions.

     Answer: Each section was ordered to the journal pattern; introduction, results and discussion, materials and method, conclusions.

 2.    Introduction. It would be very interesting that the authors of the manuscript cite some recent papers in the introduction like the year 2017 and 2018.

Answer: The recently articles (ref. 10-12, 15-16 and 27) that related to this research were added in the introduction part.

 3.    The Figures 3, 4, 5 and 6 should be clear, improve the quality of images.

Answer: All figures were adjusted to high quality and they can attach the separated figure file in the submit process.

 4.    The conclusions should be amplified and improved to clarify the results obtained in this work.

 Answer: The conclusions part was re-written with clear and concise.

Minor corrections:

Although the paper looks essentially as a valuable contribution, still some details need to be fixed, in my opinion:

1.      The English style requires improvements. The authors must be sure that there are no grammatical errors and that the manuscript is perfectly written in English.

Answer: The revised manuscript was English grammar corrected by a native speaker (Dr. Robert Douglas John Butcher, university research counselor).

2.      The authors must be sure that the bibliographical references, figures and tables are in accordance with the format of the journal.

All formats were accordance to journal requirement.

3.      It would be very interesting that the authors of the manuscript include more bibliographical references and, specially, references from the journal itself.

More related researches were collated and cited.

Round 2

Reviewer 1 Report

No further comments 

Reviewer 4 Report

I would like to recommend to check the language of the manuscript.